

# Temporal and spatial variations of atmospheric unintentional PCBs emissions in Chinese mainland from 1960 to 2019

Ye Li[1], Ye Huang[1], Yunshan Zhang[1], Wei Du[4], Shanshan Zhang[1], Tianhao He[1], Yan Li[1,3], Yan Chen[1], Fangfang Ding[1], Lin Huang[1], Haibin Xia[1], Wenjun Meng[2], Min Liu[1], Shu Tao[2]

[1]Key Laboratory of Geographic Information Science of the Ministry of Education, School of Geographic Sciences, East China Normal University, 500 Dongchuan Road, Minhang District, Shanghai 200241, China.
[2]Laboratory of Earth Surface Processes, College of Urban and Environmental Science, Peking University, Beijing 100871, China.
[3]Collaborative Innovation Center of Sustainable Forestry, College of Forestry, Nanjing Forestry University, 159 Longpan Road, Xuanwu District, Nanjing, 210037, China.
[4]Yunnan Provincial Key Laboratory of Soil Carbon Sequestration and Pollution Control, Faculty of Environmental Science & Engineering, Kunming University of Science &Technology, Kunming 650500, China.

*Correspondence to*: Ye Huang (huangye@geo.ecnu.edu.cn)

**Abstract.** Polychlorinated biphenyls (PCBs) are a group of persistent organic pollutants (POPs) that have been proven to be harmful to ecosystem and human health. Detailed information about the spatiotemporally distribution of unintentionally produced PCBs (UP-PCBs) is crucial for understanding the environmental fate and associated health risks. However, researches estimating UP-PCBs emissions in China are limited. In this study, source-specific and year-varying emission factor (EF) dataset was used to develop emission inventory of 12 dioxin-like UP-PCBs congeners covering 66 sources in Chinese mainland with spatial resolution of province for 1960 to 2019 and 0.1°×0.1° for 2019. The results indicated that historical national UP-PCBs emission represented an increasing trend until around 1995, and then showed an overall decreasing trend from 1995 to 2019. Cement production was the largest UP-PCBs emission source in Chinese mainland. Geographically, East and North China contributed largest UP-PCBs emission across Chinese mainland. High emission densities were mainly happened in the densely populated and well-developed region such as Yangtze River Delta Urban Agglomeration and Pearl River Delta. Furthermore, highly positive correlations of emission densities with population and GDP densities were identified.

## 1 Introduction

Polychlorinated biphenyls (PCBs) are a group of typical persistent organic pollutants (POPs), which have been intentionally produced for dielectric and heat exchange fluids in electronic products and unintentionally produced through a series of combustion and industrial processes (Jepson and Law, 2016; Lu et al., 2021; Zhang et al., 2013; Zhao et al., 2020). Due to characteristics of hydrophobicity and persistence, PCBs can undergo long-distance transportation and result in ubiquitous



contamination of the biosphere (Desforges et al., 2018). PCBs show toxic threat to human and animals by impairing reproduction system and disrupting endocrine and immune systems (Desforges et al., 2018; Wirgin et al., 2011). Moreover, PCBs represent biomagnification by accumulating through food web and thus pose a risk of adverse effects to wildlife

(Desforges et al., 2018; McLeod et al., 2015; Ranjbar Jafarabadi et al., 2019). In 2001, PCBs were listed as one of 12 initial POPs in Stockholm Convention, and more than 90 signatory countries committed to phasing out or eliminating large stocks or other sources of these compounds (UNEP, 2001). Globally, PCBs were mainly used commercially between 1930 and 1970s. It has been estimated that ~1.3 million metric tons (tonnes, t) were produced during this period, and mostly occurring in the United States (~48%), Russia (~13%), Germany (~12%), France (~10%), United Kingdom (~5%) and Japan (~4.4%) (Breivik

et al., 2002a, 2007). In China, ~1000 t PCBs were produced between 1965 and 1974, which was accounted for ~0.8% of the total global PCBs production (Lu et al., 2021; Xing et al., 2005). Subsequently, the use and manufacture of PCBs were banned in Japan (1972) (Kim and Masunaga, 2005), China (1974) (Zhang et al., 2013), United States (1979) (Marek et al., 2013), United Kingdom (1981) (Jurgens et al., 2015) and European Union (1987) (Jepson et al., 2016). As a result, the concentration levels of PCBs started to slowly decline in the environment around the world. However, the unintentionally produced PCBs

(UP-PCBs) are still emitted after the bans and may have become the major sources of PCBs in the current environment (Cui et al., 2013; Desforges et al., 2018; Zhao et al., 2020), posing continuous threaten to ecosystems.

Accurate estimation of PCBs emissions is essential for comprehensive understanding of their behaviors in environment and adverse effects on human health and ecosystem. Emission inventory with detailed sources, spatial-temporal information can support policy-making in mitigating PCBs pollution with cost-effective control strategies (Quaß et al., 2004) and provide

key input data for environmental modeling of PCBs (Gluge et al., 2016; ter Schure et al., 2004). The first global intentionally produced PCBs (IP-PCBs) emission inventory was introduced by Breivik et al. (2002a, b). The emission of 22 PCB congeners was estimated for 70 years in 114 individual countries with a resolution of $1° \times 1°$ (Breivik et al., 2007). On the national scale, Yang et al. (2010 ) and Cui et al. (2013) estimated the emissions of UP-PCBs in China. In their study, cement production represented the largest source for UP-PCBs emissions in China by applying emission factors (EFs) of 7.4 and 5.1 g/t based on

data in Japan (Toda, 2005; Yamamoto et al., 2011). While much lower values of 133-375 μg/t for cement kilns with fabric filter in China were reported (Liu et al., 2013), indicating that large uncertainties might still existed in emission estimation for cement production. EFs for 12 dioxin-like UP-PCBs were summered in toolkit for identification and quantification of releases of dioxins, furans and other unintentional POPs under Article 5 of the Stockholm Convention on Persistent Organic Pollutants (UNEP, 2013), but they were last updated in 2013 and lately reported EFs were not included. EFs are always with large

variations and vary with time and regions, due to differences in fuel quality, facility, abatement measures and other factors (Meng et al., 2017; Shen et al., 2013). Thus, homemade EFs were preferred (best estimate) over those reported in the toolkit in this study in order to accurate estimate emissions (Huang et al., 2014; Meng et al., 2017). Declining trends of EFs were always reported (Huang et al., 2022; Meng et al., 2017; Shen et al., 2011; Wang et al., 2012), mainly because of adoption of advanced combustion, production or abatement technologies. Technology split method which assume that the change of EFs

for a given source was due to technology transformation within the source (Bond et al., 2004) was applied to model time-





varying EFs. However, fixed EFs were applied in most of studies estimating PCBs emissions, resulting in underestimation of past emissions or overestimation of current emissions (Huang et al., 2022; Huang et al., 2014). Overall, current researches were limited by incomplete coverage of emission sources, unreliable determination of EFs and no available data for recent years (Breivik et al., 2006), resulting in large uncertainties of current popular models (Chatzikosma and Voudrias, 2007; Gluge

et al., 2016; Lei et al., 2021). Therefore, an elaborate UP-PCBs emission inventory is urgently needed.

In this study, an EF database of 12 dioxin-like UP-PCBs for various sources were compiled based on a thorough literature review (Table S1 and S2). Time-varying EFs for key sources were acquired using technology split method developed by Bond et al. (2004) and widely adopted in development of emission inventories (Meng et al., 2017; Shen et al., 2013; Wang et al., 2014a). Provincial atmospheric emissions for 12 dioxin-like UP-PCBs (Table S3) for 66 sources (Table S1) from 1960 to 2019

in Chinese mainland were compiled. Furtherly, a high spatial resolution (0.1°×0.1°) inventory for 2019 was developed based on PKU-FUEL (inventory.pku.edu.cn) (Wang et al., 2013). In addition, the uncertainties of emission estimations were quantified by Monte Carlo simulation and characteristics of geospatial and temporal variations in total, per capita emissions and emission intensities were investigated.

## 2 Methods

### 2.1 Emission sources and activity data


Various combustion and industrial processes could release PCBs. Emission factors for burn of coal, oil, biomass and waste were reported in previous studies (Lee et al., 2005; Liu et al., 2013; UNEP, 2013) and those sources were included in this study. However, there were no data available for burning of gas, these sources were thus excluded. Fuel consumption data were from PKU-FUEL (Wang et al., 2013), which has county-level energy consumption data for China and was updated to

cover the period of 1960-2019. Besides, PCBs would also emitted during the production of cement, coke, vinyl chloride monomer, iron &steel and other metals, as well as cremation of corpse. Activity data for these sources were directly acquired from China Statistical Yearbook (NBSC, 2020). Totally, 66 sources were considered in compiling the inventory and reclassified into 9 categories for further analysis (Table S1).

### 2.2 Emission factors

A thorough literature review was taken to update the EFs database by taking the recently published EFs into account. Technology split method was applied to acquire dynamic EFs. In details, coke production was classified into beehive coke and mechanical coke (NBSC, 2020); electric arc furnace, oxygen blown converter and open hearth furnace were considered for raw steel production (WSA, 2021); precalciner kiln and other kiln were taken into account for cement production (Huang et al., 2014). Regarding to applying of abatement facilities, only controlled and uncontrolled were considered due to miss of in-

depth data. This simplification has been proved to be effective in estimate emissions of dioxins (Huang et al., 2022). Choices





of EFs for each source were listed in Table S1. The composition profiles (Table S2) of 12 dioxin-like UP-PCBs for various sources were obtained to further calculate the emissions of each congener.

## 2.3 Emission estimation and spatial allocation

A bottom-up procedure was applied to compile the inventory. Provincial emissions of each PCB congener for each source

were calculated as products of the source strength and corresponding EFs. Provincial emissions of 2019 were further allocated into $0.1° \times 0.1°$ grids using various surrogates. For fuel combustion sources (including agricultural waste burning and wildfire), gridded fuel consumption data from PKU-FUEL (Wang et al., 2013) were used to disaggregate emissions. Gridded population density (Oak Ridge National Laboratory, 2020) was applied to allocate emissions from cremation of corpse. For other sources, gridded industrial coal consumption from PKU-FUEL was used as surrogate.

## 2.4 Uncertainty analysis

Monte Carlo simulations were run 100000 times considering variations both in source strength and EFs to assess the uncertainties of emission inventory. Variations of 5%, 10%, 20% and 30% were set to power&heating sector, industrial sector, residential sector and open biomass burning, respectively, with uniform distribution assumed (Huang et al., 2022; Huang et al., 2014; Shen et al., 2013; Shen et al., 2017). Although, lognormal distribution was often assumed for EFs (Shen et al., 2013;

Wang et al., 2014b), it could not directly be applied in Monte Carlo simulation in estimating emissions. Otherwise, the median value would underestimate emissions (Bond et al., 2004). On the basis of Cox's method (Zhou and Gao, 1997), the standard deviation (SD) of best estimate for EF (BE, are actually arithmetic mean instead of geometric mean) could be calculated using geometric standard deviation (σ is SD of log-transformed EFs) by:

$$\text{SD}_{BE} = \sqrt{\frac{\sigma^2}{N} + \frac{\sigma^4}{2(N-1)}} \tag{1}$$

where N is sample size. Best estimates and uncertainty ranges of emissions then could be represented by medians and interquartile ranges derived from the Monte Carlo simulations.

## 3 Results and discussion

### 3.1 National and provincial emissions from various sources in 2019

The annual total atmospheric emissions of UP-PCBs in Chinese mainland were estimated around 249.0 (interquartile range:

150.9 to 706.2) g WHO-TEQ or 1.0 (0.6-1.8) t. Total annual emissions for specific province and their source profiles are shown in Fig. 1. Hebei, Jiangsu, Shandong, Guangdong, Heilongjiang, Henan, Hubei, Hunan, and Anhui, ranked at the top, which accounted for more than 50% of total emissions (124.6 g WHO-TEQ). Our estimation of 1.1 (0.7-2.0) t in 2010 were lower than 1.5 t reported by Cui et al due to much higher EF applied for cement production in their study (Cui et al., 2013), but higher than their updated estimation of 0.7 t with lower EF applied for cement production (Cui et al., 2015), owing to wider coverage





of emission sources in this study. Liu et al. (2013) estimated emissions of 3.5 and 0.6 kg for cement and iron&steel production in 2009, respectively, which were much lower than 53.3 (24.6-116.2) and 5.2 (2.9-9.7) kg, respectively, acquired in this study. In their study, EFs were all for plants with well abatement measures, it might not be appropriate to use as best estimate for national average. Besides, they used geomean instead of arithmetic mean, which would resulted in further underestimation of emissions (Bond et al., 2004).

Among various sources, cement production (22.6%) was the largest UP-PCBs emission source in Chinese mainland in 2019, followed by industry boilers sector (17.7%) and iron and steel production (16.5%) (Fig. 1). Due to variations of development status, industrial structure, energy structure and land use cover, the source profiles showed distinct characters among different regions (Table S4) across China (Fig. 2). For example, cement production and industry boilers sector contributed most UP-PCBs emissions in Southwest China because of widespread distribution of cement plants in this area (Li

et al., 2021; NBSC, 2020; Song et al., 2019). In addition, North China accounted for more than 33% of total iron and steel production in Chinese mainland (NBSC, 2020). The contributions of iron and steel production to total emission of this region could reach 34.6%, which is much higher than that in other regions. Open biomass burning was the largest contributor (44.9%) of UP-PCBs in Northeast China due to high forest coverage and cultivation activities (Chen et al., 2019; Song and Deng, 2017) associated with high rate of wildfires and large number of crop residues burned outside (Yin et al., 2019; Zhao and Liu, 2019).

This region accounted for around 15% of wildfires (affected forest area) and 20% of crop production in 2019 in Chinese mainland, respectively (NBSC, 2020). Waste burning contributed most to UP-PCBs emissions in South China because of its abundance of e-waste sites (Chen et al., 2014; Wang et al., 2017). Non-ferrous metals were highest contributor of UP-PCBs emissions in Northwest China due to extensive distribution of non-ferrous metals plants and relatively small emissions from other sources (e.g., residential sector and waste burning due to low population density) (Huang et al., 2022).

**3.2 Composition profiles of 12 dioxin-like UP-PCBs congeners**

Fig. 3 shows composition profiles of 12 dioxin-like UP-PCBs emissions in Chinese mainland. PCB 118, PCB 105 and PCB 77 were the three largest contributors in terms of mass (Fig. 3A), which accounted for 42.1%, 30.4% and 12.2% of total emissions, respectively. Whereas PCB 126, the most toxic UP-PCBs compound, only contributed 0.2% of total emissions. The other dioxin-like UP-PCBs, including PCB 81, PCB 114, PCB 123, PCB 156, PCB 157, PCB 167, PCB 169 and PCB 189

accounted for 15.2% of total emissions. For PCB 118, PCB 105 and PCB 77, open biomass (>59%) and waste burning (>19%) were two major sources (Fig. S1). Cement production, industry boilers sector and iron & steel production contributed significantly to emission of PCB 126. Open biomass burning, waste burning and residential sector were the top three emission sources for the rest of dioxin-like UP-PCBs. In terms of TEQ (Fig. 3B), PCB 126 made the greatest contribution of 79.6%. Despite PCB 169 only accounted for 0.04% of the total emission in mass, it contributed 4.4% in terms of TEQ due to its high

toxicities. On the contrary, PCB 77, PCB 105 and PCB 118 accounted for 12.2%, 30.4% and 42.1% of total emission in mass, respectively, but reduced to 5.0%, 3.7% and 5.2% in terms of TEQ due to their relative low toxicities compared with PCB 126





and PCB 169. In general, PCB 126, PCB 169, PCB 77, PCB 105 and PCB 118 should be priority concern in aspect of protecting human health.

TEQ/u, defined as g WHO-TEQ per unit of mass of UP-PCBs emissions from a given source, was calculated to determine
the toxicities of emission sources. According to Table S5, iron and steel production was the most toxic UP-PCBs emission source with the TEQ/u value of $6.8 \times 10^{-3}$ g WHO-TEQ/g, followed by non-ferrous metals sector ($4.6 \times 10^{-3}$ g WHO-TEQ/g), others sector ($3.0 \times 10^{-3}$ g WHO-TEQ/g), cement production ($2.2 \times 10^{-3}$ g WHO-TEQ/g) and industry boilers ($1.1 \times 10^{-3}$ g WHO-TEQ/g), indicating that these sources released large amount of high toxic UP-PCB congeners. This suggests that these sources represented potential higher priority to be reduced compared with other sources in aspect of protecting human health.
However, the financial costs of reducing emissions might vary a lot among sources and regions (Wang et al., 2021), resulting in difficulty of determining sources that should be controlled in priority.

**3.3 Uneven distribution of emission densities and per-capita emissions**

Geographic distribution of UP-PCBs emission densities was shown in Fig. 2. The distribution coincides with the Huhuanyong Line, the famous empirical line that divides China into the densely populated region and sparsely populated region (Cheng et
al., 2022). High emission densities were mainly happened in the densely populated and well developed regions such as Yangtze River Delta Urban Agglomeration, Pearl River Delta, North China Plain and Sichuan Basin. The distribution was similar to other human activities related pollutants like black carbon, ammonia, PAHs, $PM_{2.5}$ and PCDD/F (Huang et al., 2022; Huang et al., 2014; Meng et al., 2017; Shen et al., 2013; Wang et al., 2014a; Wang et al., 2014b). The geographic distribution of per capita UP-PCBs emission differed from that of emission densities (Fig. S2), which showed a decrease trend from north to
south. In some sparsely populated regions of Northeast and Northwest China, such as Karamay (Xinjiang), Tsitsihar (Heilongjiang) and Golmud (Qinghai), the per capita emissions represented seriously high due to significant emissions from heavy industry with relative low population density (NBSC, 2020).

The average emission density and per capita emission of Chinese mainland were 0.03 ng WHO-TEQ/m$^2$ and 0.18 μg WHO-TEQ/person, respectively. Shanghai, Tianjin and Jiangsu represented highest emission density with values of 0.31, 0.17
and 0.15 ng WHO-TEQ/m$^2$, respectively (Fig. S3). According to Fig. 1, iron & steel production was largest contributor in these regions. The per capita emissions in these regions were below the national average mainly because that population density was extremely higher than national average (NBSC, 2020). On the contrary, Tibet, Qinghai and Xinjiang showed lowest emission density due to less developed industry and sparse population distribution. Tibet, Qinghai and Xinjiang accounted for 38% of the area, but only contributed to 2% population and 2% GDP of China in 2019 (NBSC, 2020). The relationships of
emission densities with population densities as well as GDP densities were shown in Fig. S4A and Fig. S4B. The emission densities were highly positive correlated with population ($R^2 = 0.92$) and GDP ($R^2 = 0.88$) densities. This indicates that population might be a more important driver than GDP for UP-PCBs emissions, which might be explained by that the proxy used for spatial allocation of burning non-organized small-scale waste, municipal waste and residential solid fuel use etc., which are key sources for UP-PCBs, were population instead of GDP (Wang et al., 2013).





The above results suggested that the urbanization degree (or development degree) highly impacted the distribution of UP-PCBs emissions. To better evaluate the emission disparities among regions with different levels of development, Chinese mainland was classified into city, town and rural areas based on the population density threshold method introduced by Wang et al. (2013). Fig. S5 shows emission densities and per capita emissions for city, town and rural areas from nine source categories. Despite per capita emissions followed the order of city < town < rural areas, emission densities were completely

the opposite, indicating that higher exposure risks of UP-PCBs for people living in cities than in town, and in town than in rural areas. Regarding to source profiles, the major difference was that contribution of open biomass burning was significant in rural areas but negligible in city and town.

### 3.4 Historical trends of emissions from various sectors and regions

Historical trends of UP-PCBs emissions from various source categories and regions were shown in Fig. 4. The national UP-

PCBs emissions represented an increasing trend before year around 1995. This was mainly driven by the increasing emissions from cement production during this period with cement production increased 29 times (NBSC, 2020). Then, the national UP-PCBs emissions showed an overall decreasing trend from 1995 to 2019. In this period, cement production only increased 4 times, while the EFs decreased by 94.2%. In the meantime, industrial consumption of coal, production of iron and steel increased by factors of 1.5, 7.6 and 8.6, respectively. However, their EFs decreased by 56.6%, 74.8% and 45.5%, respectively.

Besides, UP-PCBs emissions from waste burning, non-ferrous metal sector, residential sector and power & heating sector were kept increasing before 1990 but decreasing ever since. This trend was similar with those for pollutants like black carbon, $PM_{2.5}$, PAHs and PCDD/Fs (Huang et al., 2022; Huang et al., 2014; Shen et al., 2013; Wang et al., 2012), mainly because of widely applying abatement facilities (Shen et al., 2021). The variation of residential sector was mainly resulted from energy structures transition (Tao et al., 2018). It should be pointed out that the historical emissions might be associated with large uncertainties

due to lack of EF data in early years and in-depth information about technology split, as EFs for a given source could vary multiple orders of magnitude (UNEP, 2013).

Fig. 4A shows temporal trends of emissions from different regions in Chinese mainland. Cement production was the major contributor across Chinese mainland from 1960 to 2019, followed by waste burning and industry boilers sector. In general, cement production emissions were increasing from 1960 to around 1995 except for Northwest and Northeast China.

From 1995 to 2019, the contribution of iron & steel sector increased a lot in North China as production of iron & steel increased by about 13 times. In the same period, the emission contribution from industry boilers sector was substantial increased in East, Central, Southwest and Northeast China (Fig. S6). Waste burning showed an overall decreasing trend from 1960 to 2019 at the national scale owing to continuous reducing EF required by the more and more strict emission standard (Cheng et al., 2020). In Northeast China, open biomass burning was significant increased from 2014 to 2019, which might be explained by

that the increasing temperature resulted in more and more intensive wildfires (Diffenbaugh et al., 2021; Running, 2006).



## 3.5 Spatial-temporal characteristics of emission intensity

Emission intensity (EI), defined as anthropogenic emissions per unit of GDP, were calculated. EIs were with great disparities in space and varied over time (Fig. 5). In 1960, high EIs (>2000 pg WHO TEQ/Yuan) were mainly found in central of China, including Gansu, Shanxi and Sichuan. Besides, Liaoning and Zhejiang were also listed in the provinces with high EIs. Except

Zhejiang, these provinces shared the same largest source, which is cement production. Their contributions for national cement production were 1.3-3.2 times of those for GDP. While high EIs of Zhejiang was likely caused by intensified municipal waste burning (NBSC, 2020). In 2019, provinces with relatively high EIs (>5 pg WHO TEQ/Yuan) were mainly located in north of China, including Heilongjiang, Ningxia, Inner Mongoria, Shanxi and Hebei. High EIs for these provinces were owing to relative larger production of iron & steel, cement or coal consumption per unit of GDP. The overall EI for Chinese mainland

decreased from 1480 pg WHO TEQ/Yuan in 1960 to 2.5 pg WHO TEQ/Yuan in 2019 (Fig. S7). Continuous decreasing trends of EI for all regions could also be observed (Fig. S7). These might be explained by the increasing contribution of GDP from tertiary industry (NBSC, 2020), which associated with relative low emissions. Besides, increasing adoption rate of abatement measures driving by the more and more strict emission standards (Huang et al., 2014) might have also contributed to these decreasing trends. Furtherly, the relationship between EIs and per capita GDP was investigated (Fig. S8). EIs showed a

decreasing trend with the increase of per capita GDP. The decreasing rate was much faster when per capita GDP was less than about 20-thousand Yuan/person (Fig. S8A). If only looking at the recent data, $R^2$ could reach 0.72 (Fig. S8B). This suggested that per capita GDP might use a predictor for EIs.

## 4 Conclusions

Information about historical trends and spatial distribution of UP-PCBs in China remains scarce. In this study, source-specific

and yearly varying EF dataset was used to develop the emission inventory for 12 dioxin-like UP-PCBs congeners in Chinese mainland from 1960 to 2019 with 66 sources covered. 0.1°×0.1° gridded inventory for 2019 was further compiled. This study showed that total emissions of UP-PCBs in Chinese mainland were around 249 g WHO-TEQ in 2019. Cement production, industry boiler sector and iron & steel production were three major sources. Historically, national UP-PCBs emission showed increasing trend from 1960 to 1995, and subsequently decreasing from 1995 to 2019. Geographically, North and East China

were the largest contributors of UP-PCBs emission across Chinese mainland, which indicated that high emission of UP-PCBs mainly happened in densely populated and developed regions. For specific congeners, PCB 118, 105 and 77 were major UP-PCBs in terms of mass, while PCB 126, 77 and 169 were major UP-PCBs in terms of g WHO-TEQ. These UP-PCBs congeners above should be concerned with priority in aspect of protecting human health and mitigating adverse effects to ecosystem. TEQ/u of each source was calculated to determine the toxicities of emission sources. Iron & steel production, non-ferrous

metals sector, cement production and industry boilers were with highest TEQ/u and might be controlled in priority. The emission densities were highly correlated with population densities and GDP densities, indicating that urbanization degree highly impacted the distribution of UP-PCBs emissions.



Large uncertainties still exit in current emission inventory due to insufficient EF data (e.g. source emission profile) and lack of in-depth source information (e.g. geolocation of point source). Despite technology split method was applied in this study, only "controlled" or "not controlled" were included in this study considering the availability of relevant data, resulting in uncertainties of modeled trends of EFs with time. Future works might focus on acquiring detailed home-made EFs and developing more accurate method in allocate emission across space and time to estimate emissions more realistic and reduce temporal and spatial biases.

**Author contributions**

The study was completed with contribution from all authors. Ye Huang, Min Liu and Ye Li designed the research; Ye Huang, Ye Li, Yunshan Zhang and Shanshan Zhang conducted the experiment; Wei Du, Tianhao He, Yan Li, Yan Chen, Fangfang Ding, Lin Huang, Haibin Xia and Wenjun Meng analyzed experimental results; Ye Li and Ye Huang wrote the paper; Min Liu and Shu Tao supervised the research.

**Competing Interests**

The authors declare that they have no conflict of interest.

**Code/Data availability**

Code and data used in this article are available by contacting the corresponding author.

**Acknowledgements**

The present study is funded by National Natural Science Foundation of China (No. 41730646, 42230505, 41907313, 42277388, 42206148), National Key R&D Program of China (No. 2020YFC1806700), Science and Technology Commission of Shanghai Municipality (No. 19ZR1415100), China Postdoctoral Science Foundation (No. 2020M671047) and National Social Science Foundation of China (No. 20BRK022).

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





**Figures**

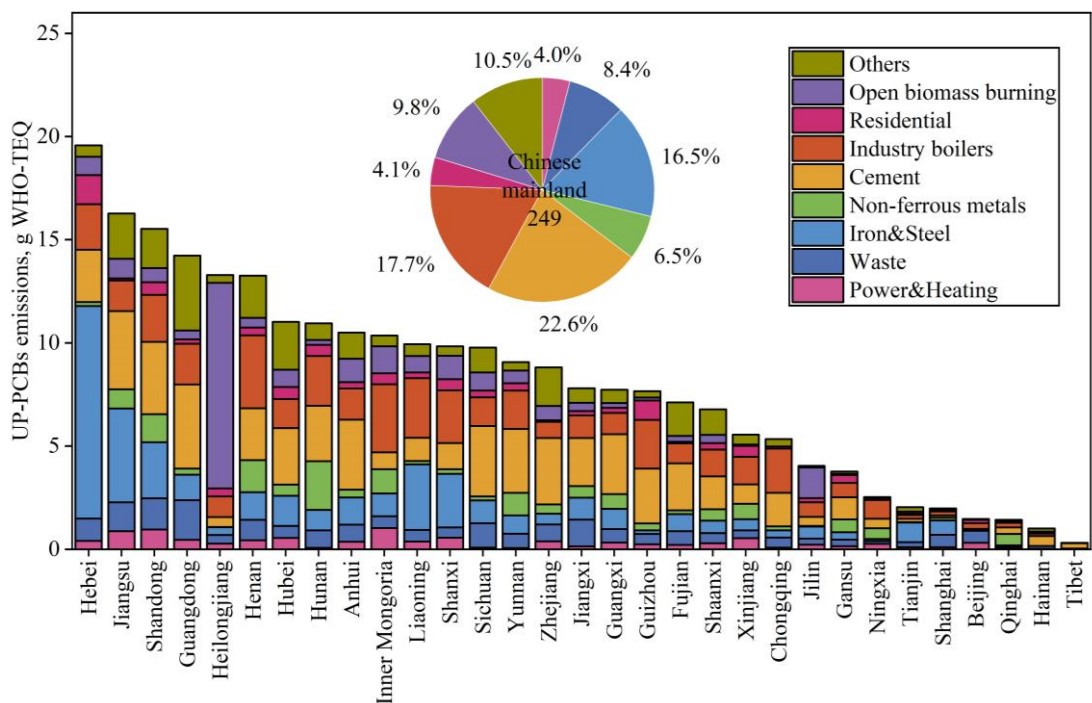

**Fig. 1. Total (pie chart) and provincial (columns) emissions for Chinese mainland in 2019.**






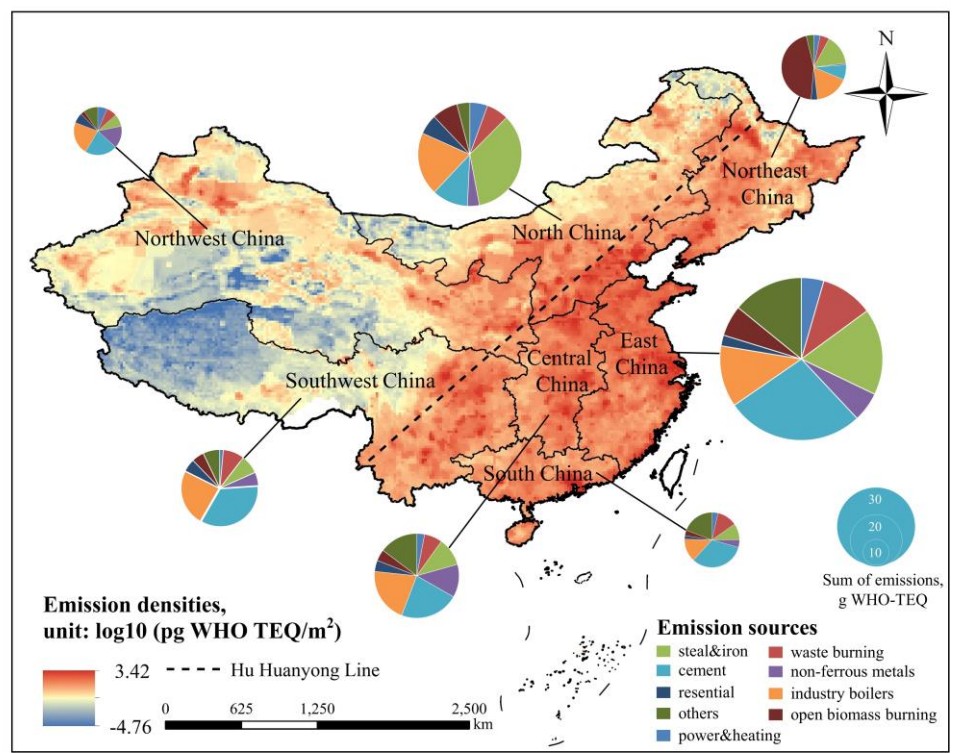

**Fig. 2. Spatial distribution of emission densities across Chinese mainland (the resolution is 0.1°×0.1°) and sum of emissions from various emission sources in different regions in 2019.**


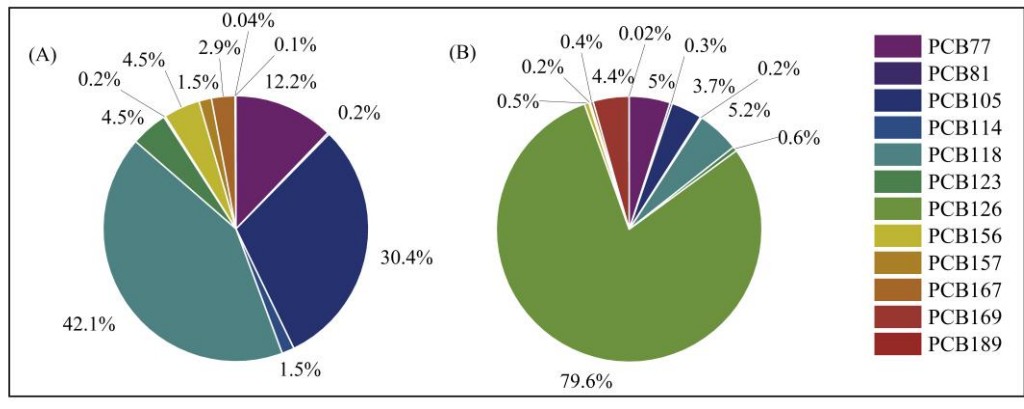

**Fig. 3. Emission profiles for 12 dioxin-like UP-PCBs in Chinese mainland in 2019.** Pie chart for PCBs congeners in terms of mass (A) and TEQ (B), respectively.




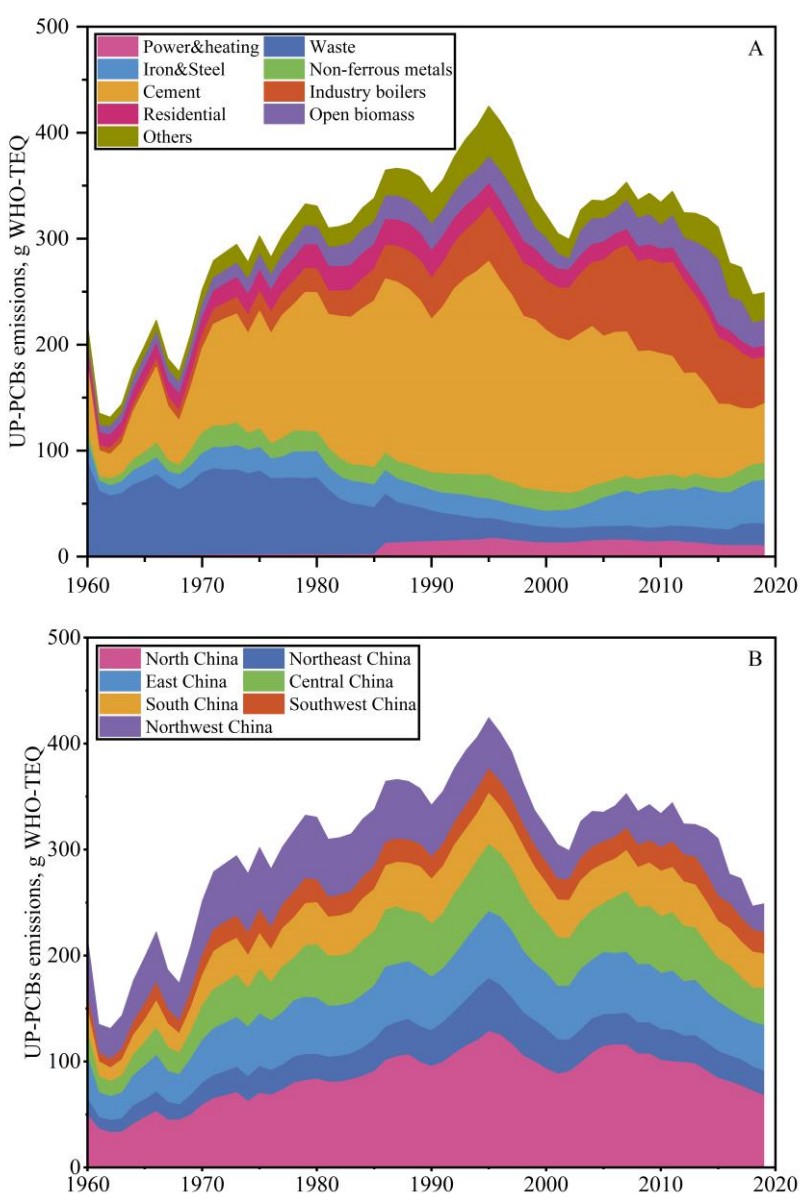

**Fig. 4. Temporal variations of UP-PCBs for different sources (A) and regions (B).**





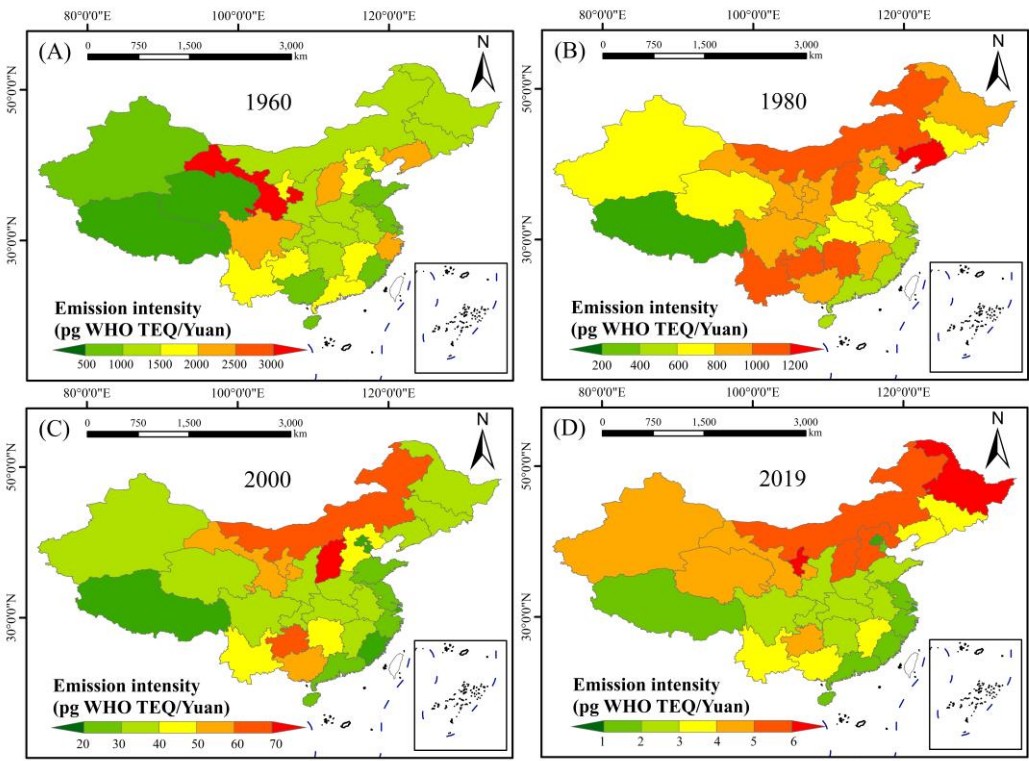

**Fig. 5. Emission intensities of UP-PCBs (pg WHO TEQ/Yuan) across Chinese mainland in the year of 1960 (A), 1980 (B), 2000 (C) and 2019 (D).**