# Peer review of "Temporal and spatial variations of atmospheric unintentional PCBs emissions in Chinese mainland from 1960 to 2019"

_EGUsphere, 2022_

## Referee Comment (RC1)

This manuscript developed the emission inventory in Chinese mainland for 12 dioxin-like UP-PCBs congeners from 66 sources with a resolution of 0.1°×0.1°from 1960 to 2019. The characteristics of geospatial and temporal variations in total, per capita emissions and emission intensities were also investigated. This UP-PCBs emissions provides an essential data support for the assessment of human health risks from exposure to PCBs and for policy-makers to optimize PCBs mitigation strategies, and key input data for environmental modeling of PCBs. In general, the paper is well written, although some places require modification and clarification. It can be published after minor revisions.

Specific comments
1. This study established emissions inventories of 12 dioxin-like UP-PCBs for various sources by using the economic activities and corresponding EFs. But the EFs listed in Table S1 are total EFs for the sum of 12 dioxin-like UP-PCBs. How can obtain the EF for each UP-PCB from various sources? The detail EFs for individual dioxin-like UP-PCB should be presented.

2. In the Section 2.3, for other sources, gridded industrial coal consumption from PKU-FUEL was used as surrogate. This surrogate data for spatial allocation is not appropriate for some emission sources. I suggest gridded field fire data can be used to allocate the emissions from open biomass source. The surrogate data for waste incineration should use the spatial distribution of incinerator in different provinces.

3. Line 148, Fig. 3 shows UP-PCBs emission profiles for 9 source categories in Chinese mainland. But Fig. 3 shows the emission profiles for 12 dioxin-like UP-PCBs. The results (line 152-155) for emission profiles for 12 UP-PCBs from various sources also can not be seen from Fig. 3. The authors should add one figure to show the 12 UP-PCBs emission profiles for 9 source categories.

4. Line 214-215, Cement production was the 215 major contributor across Chinese mainland from 1960 to 2019, followed by waste burning and industry boilers sector. This result and be seen from Fig. 4A, not Fig. S5.

5. The spatial resolution of Fig. 2 should be given.

6. Line 50, H. et al., 2004? Please check it.

7. Line 105. "Total population" should be gridded population density, and the data source should be given.

8. The representative meaning of σ in eq. 1 should be given.

9. The year should be given in Fig.2 and Fig. 3.

10. The unit of y-axis in Fig. S5 should be given.

---

## Author Comment (AC1)

**Response to Reviewer**

Dear Reviewer:

Thanks for your comments concerning our manuscript entitled "**Temporal and spatial variations of atmospheric unintentional PCBs emissions in Chinese mainland from 1960 to 2019**" (Ms. Ref. No.: EGUSPHERE-2022-977). Those comments are very valuable and helpful in improving our paper. We have studied the comments carefully and made corrections correspondingly. The main corrections in the paper and our point-by-point responses to reviewer's comments are presented below.

**Responses to Reviewer #2**

**General comments:** This manuscript developed the emission inventory in Chinese mainland for 12 dioxin-like UP-PCBs congeners from 66 sources with a resolution of 0.1°×0.1°from 1960 to 2019. The characteristics of geospatial and temporal variations in total, per capita emissions and emission intensities were also investigated. This UP-PCBs emissions provides an essential data support for the assessment of human health risks from exposure to PCBs and for policy-makers to optimize PCBs mitigation strategies, and key input data for environmental modeling of PCBs. In general, the paper is well written, although some places require modification and clarification. It can be published after minor revisions.

**Response:** We really appreciate your valuable comments and suggestions. We have made corrections correspondingly according to your comments. Please check point to point response as below.

**Comment 1.** This study established emissions inventories of 12 dioxin-like UP-PCBs for various sources by using the economic activities and corresponding EFs. But the EFs listed in Table S1 are total EFs for the sum of 12 dioxin-like UP-PCBs. How can obtain the EF for each UP-PCB from various sources? The detail EFs for individual dioxin-like UPPCB should be presented.

**Response:** To estimate emissions for each PCB congener, emission profiles of 12 dioxin-like UP-PCBs were acquired by a thorough literature review. For source without data, emission profile of similar source was applied. Table S2 for final choices of emission profile for each source and corresponding references can be seen in supplement.

**Comment 2.** In the Section 2.3, for other sources, gridded industrial coal consumption from PKUFUEL was used as surrogate. This surrogate data for spatial allocation is not appropriate for some emission sources. I suggest gridded field fire data can be used to

allocate the emissions from open biomass source. The surrogate data for waste incineration should use the spatial distribution of incinerator in different provinces.

**Response:** Open biomass including wildfire and agricultural waste burning were included in PKU-FUEL. The proxy used for allocation of these sources in PKU-FUEL was actually gridded field fire data from GFED v4. We made corresponding description as below.

"For fuel combustion sources (including agricultural waste burning and wildfire), gridded fuel consumption data from PKU-FUEL (Wang et al., 2013) were used to disaggregate emissions."

However, information about spatial distribution of incinerator was hard to acquire, so industrial coal was used as proxy to disaggregate emission in space in current inventory and also in PKU-FUEL. We noted that this would result in spatial bias of inventory. We made corresponding description in conclusion section.

Line 253-254. "Large uncertainties still exit in current emission inventory due to insufficient EF data (e.g. source emission profile) and lack of in-depth source information (e.g. geolocation of point source)."

Line 256-258. "Future works might focus on acquiring detailed home-made EFs and developing more accurate method in allocate emission across space and time to estimate emissions more realistic and reduce temporal and spatial biases."

**Comment 3.** Line 148, Fig. 3 shows UP-PCBs emission profiles for 9 source categories in Chinese mainland. But Fig. 3 shows the emission profiles for 12 dioxin-like UP-PCBs. The results (line 152-155) for emission profiles for 12 UP-PCBs from various sources also can not be seen from Fig. 3. The authors should add one figure to show the 12 UP-PCBs emission profiles for 9 source categories.

**Response:** Sorry for the mistake and it was revised as "Fig. 3 shows composition profiles of 12 dioxin-like UP-PCBs emissions in Chinese mainland." In addition, we added one figure showing emission profiles for 9 source categories (Fig. S1) and a table of detailed composition profile for each source (Table S2).

**Comment 4.** Line 214-215, Cement production was the 215 major contributor across Chinese mainland from 1960 to 2019, followed by waste burning and industry boilers sector. This result and be seen from Fig. 4A, not Fig. S5.

**Response:** We apologize for this careless citation. We revised this sentence as your comment.

"Fig. 4A shows temporal trends of emissions from different regions in Chinese mainland. Cement production was the major contributor across Chinese mainland from 1960 to 2019, followed by waste burning and industry boilers sector. In general, cement production emissions were increasing from 1960 to around 1995 except for Northwest and Northeast China. From 1995 to 2019, the contribution of iron & steel sector increased a lot in North China. In the same period, the emission contribution from industry boilers sector was substantial increased in East, Central, Southwest and Northeast China (Fig. S6)."

**Comment 5.** The spatial resolution of Fig. 2 should be given.

**Response:** Thank you for the comment. The spatial resolution of Fig. 2 was 0.1°×0.1°. We added it in the caption of Fig. 2.

"Fig. 2. Spatial distribution of emission densies across Chinese mainland (the resolution is 0.1°×0.1°) and sum of emissions from various emission sources in different regions in 2019."

**Comment 6.** Line 50, H. et al., 2004? Please check it.

**Response:** We apologize for this incorrect reference. The correct reference was revised in this sentence.

"key input data for environmental modeling of PCHs (Gluge et al., 2016; ter Schure et al., 2004)"

**Comment 7.** Line 105. "Total population" should be gridded population density, and the data source should be given.

**Response:** "Total population" has been replaced with "Gridded population density".

"Gridded population density (Oak Ridge National Laboratory, 2020)"

**Comment 8.** The representative meaning of σ in eq. 1 should be given.

**Response:** σ is the standard deviation of log-transformed EFs. Explanation for meaning of σ was added in the main text accordingly.

"σ is SD of log-transformed EFs"

**Comment 9.** The year should be given in Fig.2 and Fig. 3.

**Response:** The captions of the two figures were revised as follows.

"Fig. 2. Spatial distribution of emission densities across Chinese mainland (the resolution is 0.1°×0.1°) and sum of emissions from various emission sources in different regions in 2019."

"Fig. 3. Emission profiles for 12 dioxin-like UP-PCBs in Chinese mainland in 2019. Pie chart for PCBs congeners in terms of mass (A) and TEQ (B), respectively."

**Comment 10.** The unit of y-axis in Fig. S5 should be given.

**Response:** The title (UP-PCBs emissions, g WHO-TEQ) for y axis was added for Fig. S5 (now is Fig. S6). Please check Fig. S6 in supplement.

---

## Author Comment (AC2)

**Response to Reviewer**

Dear Reviewer:

Thanks for your comments concerning our manuscript entitled "**Temporal and spatial variations of atmospheric unintentional PCBs emissions in Chinese mainland from 1960 to 2019**" (Ms. Ref. No.: EGUSPHERE-2022-977). Those comments are very valuable and helpful in improving our paper. We have studied the comments carefully and made corrections correspondingly. The main corrections in the paper and our point-by-point responses to reviewer's comments are presented below.

**Responses to Reviewer #1**

**General comments:** This manuscript presents an estimation on emissions of 12 dioxin like UP-PCBs from China. Historical provincial emissions from 66 sources in China were estimated. Furthermore, a 0.1º × 0.1º gridded emission inventory for 2019 was developed. This study provides key information about temporal and spatial characteristics of UP-PCBs emissions in China. The inventory could be used as key input in modelling the environment behaviours of UP-PCBs and support the assessment of health and ecology risks due to UP-PCBs pollution. The paper is overall well organized and properly documented. It can be published after minor revisions.

**Response:** We really appreciate your valuable comments and suggestions. We have made corrections correspondingly according to your comments. Please check point to point response as below.

**Comment 1.** PCBs have many congeners, but only 12 congeners were considered in this study, a brief explanation of choosing these 12 congeners might add in the introduction.

**Response:** Thank you for the comment. Our study mainly focused on unintentionally produced dioxin like PCBs. In 2005, WHO evaluated the toxic equivalence factors of dioxins and dioxin-like compounds and 12 PCB congeners were identified with potential significant adverse impacts on human health and environment. Also, most of studies only focused on these 12 PCB congeners. As a result, emission factors for PCBs other than these 12 PCB congeners were scarce, making it hard to estimate their emissions. Thus, 12 dioxin-like PCBs were included in this study. The description of 12 PCBs congeners was added in Line 37 in preprint.

"other sources of these compounds (UNEP, 2001). In 2005, WHO evaluated the toxic equivalence factors of dioxins and dioxin-like compounds and 12 PCB congeners were identified with potential significant adverse impacts on human health and environment. Globally, PCBs were mainly used commercially".

**Comment 2.** The first sentence in the abstract is about PCBs, but the following contents are about dioxin like UP-PCBs. This sentence should be reorganized.

**Response:** Thanks for the comment. The sentence was revised as "Polychlorinated biphenyls (PCBs) are mainly unintentionally produced with the ban of their use and manufacture, and they, especially those dioxin-like compounds have been proven to be harmful to ecosystem and human health."

**Comment 3.** In section "2.3 Emission estimation and spatial allocation", spatial allocation of fuels in PKU-FUEL might be briefly described here

**Response:** Thank you for the comment. This section was revised as: "A bottom-up procedure was applied to compile the inventory. Provincial emissions of each PCB congener for each source were calculated as products of the source strength and corresponding EFs. Provincial emissions of 2019 were further allocated into $0.1° \times 0.1°$ grids using various surrogates. For fuel combustion sources (including agricultural waste burning and wildfire), gridded fuel consumption data from PKU-FUEL (Wang et al., 2013) were used to disaggregate emissions. In PKU-FUEL, county level fuel consumption data were acquired and were further disaggregated into $0.1°×0.1°$ grids using GDP, rural population, urban population or total population depending on fuel categories. For fuel consumed in power plants and production of lime, coke, as well as aluminum, the geolocations of the sources were obtained to allocate emissions. For agricultural waste burning and wildfire, emissions were firstly allocated into $0.25°×0.25°$ grids based on dry matter burned from GFED4 (van der Werf et al., 2017) and were further disaggregated into g $0.1° \times 0.1°$ using vegetation density (Friedl et al., 2002). For cremation of corpse, which was a PCBs source included in this study, gridded population density (Oak Ridge National Laboratory, 2020) was applied to allocate emissions. For other sources that were not included in PKUE-FUEL, gridded industrial coal consumption from PKU-FUEL was used as surrogate."

Van der Werf, G., Randerson, J., Giglio, L., van Leeuwen, T., Chen, Y., Rogers, B., Mu, M., van Marle, M., Morton, D., Collatze, G., Yokelson, R.: Global fire emissions estimates during 1997–2016, Earth Syst. Sci. Data, 9, 697-720, https://doi.org/10.5194/essd-9-697-2017, 2017.

Friedl, M.A., McIver, D.K., Hodges, J.C.F., Zhang, X.Y., Muchoney, D., Strahler, A.H., Woodcock, C.E., Gopal, S., Schneider, A., Cooper, A., Baccini, A., Gao, F., Schaaf, C.: Global land cover mapping from MODIS: algorithms and early results, Remote Sens. Environ., 83, 287-302, https://doi.org/10.1016/S0034-4257(02)00078-0, 2002.

**Comment 4.** Line 66: resulting in potential underestimation of

**Response:** Thank you for the comment. This sentence has been revised to "resulting in underestimation of".

**Comment 5.** Line 74: Provincial atmospheric emissions of 12 dioxin-like UP-PCBs.

**Response:** Thank you for the comment. This sentence has been revised to "Provincial atmospheric emissions of 12 dioxin-like UP-PCBs".

**Comment 6.** Line 94: due to lack of in-depth data.

**Response:** Thank you for the comment. This sentence has been revised to "due to lack of in-depth data".

**Comment 7.** Line 131: Due to variations in
**Comment 8.** Line 132: land use/cover

**Response:** Thank you for the comment. This sentence has been written to "Due to variations in development status, industrial structure, energy structure and landuse/cover,".

**Comment 9.** Line 157: should be the priority concern

**Response:** Thanks for your comment. This sentence has been revised to "should be the priority concern".

**Comment 10.** Line 176: change "represented" to "were"

**Response:** Thanks for your comment. This sentence has been revised to "the per capita emissions were seriously high".

**Comment 11.** Line 182: change "extremely" to "much"

**Response:** Thanks for your comment. This sentence has been revised to "was much higher".

**Comment 12.** Line 205-206: were rising until 1990, but have been decreasing ever since

**Comment 13.** Line 206: was similar to that of other pollutants

**Response:** Thank you for the comment. This sentence has been written to "This trend was similar to that of other pollutants".

**Comment 14.** Line 224-226 is not clear, should rewrite

**Response:** The two sentences were revised as "For these provinces except Zhejiang, cement production was the largest source and their relative contributions to national cement production were 1.3-3.2 times of those for national GDP."

**Comment 15.** Line 245: "which indicated … regions" should be removed as there is no necessary connection

**Response:** Thanks for your comment. This sentence was removed.

**Comment 16.** Line 253: exist

**Response:** We apologize for this careless writing. This sentence was revised to "Large uncertainties still exist in current emission inventory".

**Comment 17.** In Fig. S6, "UP-PCBs emissions, g WHO-TEQ" in blue should be removed

**Response:** We apologize for this layout. This sentence was removed in supplement.